# A Review of Endothelium-Dependent and -Independent Vasodilation Induced by Phytochemicals in Isolated Rat Aorta

**DOI:** 10.3390/ani9090623

**Published:** 2019-08-29

**Authors:** Marcela Knox, Raúl Vinet, Lida Fuentes, Bernardo Morales, José L. Martínez

**Affiliations:** 1Laboratory of Pharmacology, Center of Micro Bioinnovation (CMBi), Faculty of Pharmacy, Universidad de Valparaíso, Valparaíso 2360102, Chile; 2Centro Regional de Estudios en Alimentos Saludables (CREAS), CONICYT-GORE (Proyecto R17A10001), Valparaíso 2362696, Chile; 3Faculty of Chemistry and Biology, Universidad de Santiago de Chile, Santiago 9160020, Chile; 4Vice Chancellor of Investigation, Development and Innovation, Universidad de Santiago de Chile, Santiago 9160020, Chile

**Keywords:** rat aorta, phytochemical, plant extract, vasodilation, relaxation, endothelium, endothelial cells, vascular smooth muscle cells, vessels

## Abstract

**Simple Summary:**

Cardiovascular diseases are the leading cause of death worldwide, so the investigation of new therapeutic tools is a priority for their prevention and treatment. This review shows the relevant contribution of the isolated rat aorta as an in vitro experimental model to validate the therapeutic potential of phytochemicals, mainly those present in plants traditionally used in folk medicine to relieve hypertension. The results of the assays carried out in this model show that a variety of plant extracts and their isolated compounds produce vasodilation, which may explain their use, especially to treat hypertension.

**Abstract:**

This review discusses the contribution of the use of the isolated rat aorta (IRA) as a model for the evaluation of extracts and metabolites produced by plants with a vasodilator effect in animals. This model continues to be a valuable approach for the search and development of new phytochemicals consumed as medicinal plants or foods. In most cases, the sources of phytochemicals have been used in folk medicine to treat ailments that include hypertension. In this model, the endothelium is emphasized as a key component that modulates the vessel contractility, and therefore the basal tone and blood pressure. Based on the functional nature of the model, we focused on studies that determined the endothelium-dependent and -independent vasodilatory activity of phytochemicals. We describe the mechanisms that account for aorta contraction and relaxation, and subsequently show the vasoactive effect of a series of phytochemicals acting as vasodilators and its endothelium dependence. We highlight information regarding the cardiovascular benefits of phytochemicals, especially their potential antihypertensive effect. On this basis, we discuss the advantages of the IRA as a predictive model to support the research and development of new drugs that may be of help in the prevention and treatment of cardiovascular diseases, the number one cause of death worldwide.

## 1. Introduction

In 1980, Robert F. Furchgott showed for the first time that the presence of vascular endothelium was essential for the acetylcholine (ACh)-mediate vasodilation in the isolated aorta [1]. Nowadays, it is well established that nitric oxide (NO) released by the endothelium is mainly responsible for the relaxation induced by ACh [2,3]. This discovery has helped pave the way for a deeper understanding of the inter and intracellular vascular communication that underpins the regulation of regional perfusion to meet local oxygen demand in different tissues [4,5].

The primary function of the circulation is to supply oxygen and nutrients to tissues while eliminating metabolic waste. Thus, the animal organism has an extensive vascular network composed of blood vessels, which include arteries, arterioles, capillaries, venules, and veins to accomplish these tasks. Due to its high basal oxygen uptake, the heart and the brain are particularly sensitive to the adjustments in perfusion, so have multiple mechanisms for the fine-tuning of vascular tone [6]. The dependence of the heart and brain on oxygen and nutrient supply as well as the high incidence of cardiovascular diseases explain why ischemic heart disease and stroke are the leading causes of death worldwide, with 15.2 million deaths in 2016 [7].

Regarding cardiovascular diseases, hypertension is the main factor contributing to premature death worldwide and accounted for almost 10 million deaths in 2015 [8]. Hypertension is also a significant risk factor for heart failure, chronic kidney disease, and cognitive impairment. It is estimated that over one billion people have hypertension; as the population ages and adopts more sedentary lifestyles, the global prevalence of hypertension will continue to rise to reach 1500 million by 2025 [8].

In the 1960s, hypertension was considered essentially as a disease and not as a treatable condition [9]. In 1964, the first multicenter cooperative study for diastolic hypertension, which ranged between 90- and 129-mm Hg, was started and was completed in 1971, establishing for the first time that the treatment of diastolic hypertension reduced cardiovascular events [9]. Isolated systolic hypertension was not considered treatable until 1991 when the benefits of treating it using a simple regimen with chlorthalidone (a diuretic) and adding atenolol (an α_1_-adrenergic selective blocker) were shown. In the following decades, several studies have been carried out to test the use of different classes and combinations of antihypertensive drugs [9]. Pharmacotherapy can effectively control blood pressure; however, all drugs can produce adverse effects including liver and kidney dysfunction and the economic burden of long-term medication can lead to non-compliance in therapy. The above could explain why more and more people choose non-drug therapies to treat hypertension [8,10].

The American College of Cardiology/American Heart Association recommends that patients adopt non-pharmacological interventions such as weight control, healthy habits with a reduced intake of sodium in the diet, and active physical exercise in the early stage of hypertension [11]. Thus, in many countries around the world, patients frequently consume phytomedicine and alternative medicine to improve the therapeutic effect of conventional drugs. The interest in herbal medicine can be explained by the growing amount of information showing that many phytochemicals, particularly polyphenolic compounds, have healthy properties including benefits for cardiovascular health [12,13].

Preclinical bioactivity assays require appropriate models that allow the determination and projection of the fundamental pharmacodynamic parameters, necessary for the clinical phase of the development of new drugs. In this sense, the isolated rat aorta (IRA) model has been used extensively as a standard model to evaluate vasoactive substances. Its usefulness is based on its advantages over other in vitro experimental models that seek to develop new vasoactive therapeutic tools. In this regard, we can highlight its intermediate biological complexity between the cellular level and the animal model, the reproducibility of the contractile response, and the rich counteracting signals between vascular smooth muscle cells (VSMCs) and endothelial cells (ECs) [14]. Furthermore, a large body of knowledge has accumulated through years of studies using the IRA model. A PubMed search (July 2019) using the term “isolated aorta” resulted in 14.712 articles where those performed in rats represented 43%, followed by humans (38%), rabbits (16%), mice (11%), pigs (8%), and bovines (7%). Interestingly, when the search added the term “mechanism”, the results led to 2709 articles with those carried out in rats representing 61% (humans 24%, rabbits 16%, mice 15%, pig 7%, and bovine 6%). These results clearly show the relevance of the isolated aorta model obtained from rats in biomedical research where the annual productivity has remained almost constant from 1990 until now.

## 2. Materials and Methods

We carried out a systematic review of the scientific literature using the PubMed database accessed in March 2019. The PubMed search was performed using the following MeSH (Medical Subject Headings) terms: “Aorta, Thoracic”, accounting for 33,263 search results (413 in 1981 and peaked at 1015 in 2013); “Phytochemicals”, accounting for 11,013 search results (36 in 1981 and peaked at 1134 in 2018); and “Polyphenols”, which accounted for 17,428 search results (four in 1981 and peaked at 1532 in 2017). The selected scientific works were only those that fulfilled both the search and methodological criteria. The research inclusion criterion was that all the extracts and compounds evaluated for their vasoactive potential followed a suitable method for the preparation of the IRA.

### 2.1. Isolated Rat Aortic Preparation

Proper preparation is essential to obtain and project the pharmacodynamic parameters of the substances that are tested in any model. Here, we briefly describe the experimental protocol for obtaining an isolated rat aorta preparation that allows for the estimation of reliable bioactivity data [14,15].

Rats weighing 220–280 g are sacrificed by an overdose of CO_2_ inhalation. The thoracic aorta is carefully excised and placed in a Petri dish containing modified Krebs–Henseleit buffer (KHB) (in mM): NaCl 122; KCl 4.7; NaHCO_3_ 15.5; KH_2_PO_4_ 1.2; MgCl_2_ 1.2; CaCl_2_ 2.0; D-glucose 11.5; EDTA 0.026; pH 7.4. The aorta is dissected, cleaned of connective tissue, and divided into rings segments (4–5 mm) that are suspended between two L-shaped stainless-steel hooks and placed into a 20–30 mL organ chambers containing modified KHB at 37 °C and continuously oxygenated with carbogen (95% O_2_, 5% CO_2_). Isometric tensions are measured using a force-displacement transducer connected to a polygraph. The rings are exposed to basal tension (1.5–2.0 g) for 60 min. Then, the rings are progressively stretched with a depolarizing modified KHB with high K^+^ (we used 70 mM K^+^) until a maximum stable contraction (maximal tension) is reached. Rings are repeatedly washed and equilibrated for 30 min. The functional integrity of the endothelium can be assessed by the endothelium-dependent relaxation induced by 1 μM ACh in 0.1 μM phenylephrine (PHE)-precontracted rings. At this point, the preparation is ready to evaluate the changes in aortic contractility (i.e., relaxation or contraction) against the compound of interest. The analysis of bioactivity data includes the estimation of the maximal response and the concentration that causes 50% of the maximal response (EC_50_), which can be obtained by nonlinear regression with a four-parameter logistic equation (also known as the Hill equation).

### 2.2. The Contractility Mechanism in Vessels

Figure 1 shows a simplified representation of the interaction between VSMCs and ECs to support the analysis and comprehension of phytochemical activity tested in the IRA model. Regardless of its etiology, hypertension is determined by an increase in vascular resistance that leads to an increase in blood pressure. Peripheral resistance is a function of arterial diameter, which in turn is associated with the contractility state (vasomotor tone) of VSMCs [16]. Contraction of VSMCs occur through an electromechanical and chemical–mechanical coupling, or a combination of both. The first mechanism of contraction is initiated by a change in the membrane potential, while the second type is mediated by G-protein-coupled receptors (GPCRs) and the subsequent generation of second messengers; however, both mechanisms lead to Ca^2+^-dependent activation of the myosin light chain kinase and actomyosin cross-bridges associated with structural changes and force generation [17,18].

K^+^ channels play an essential role in controlling the membrane potential in all cell types, particularly in excitable cells like VSMCs [19,20]. In VSMCs, K^+^ channels, through the control of membrane potential, determine the activity of voltage-dependent Ca^2+^ channels (VDCCs), which are the main gateway for Ca^2+^ entry into the cells. Thus, the closing of K^+^ channels results in membrane depolarization, allowing the activation of VDCCs and entry of extracellular Ca^2+^, which can be accompanied by the release of Ca^2+^ from intracellular Ca^2+^ stores such as sarcoplasmic reticulum and mitochondria. The result is an increase in the concentration of cytosolic Ca^2+^, allowing VSMC contraction and, therefore, vasoconstriction. Conversely, the opening of K^+^ channels causes membrane hyperpolarization and the closing of VDCCs. The concentration of cytosolic Ca^2+^ is reduced by the plasma membrane Ca^2+^-ATPase (PMCA) and Na/Ca exchanger (NCX), which transport Ca^2+^ outside the cell, while the Ca^2+^-ATPase in the sarcoplasmic reticulum and mitochondria recover Ca^2+^ into these stores. In this case, the result is a decrease in the concentration of cytosolic Ca^2+^, causing VSMC relaxation and consequently, vasodilation [21].

GPCRs are activated by a wide variety of stimuli including the largest family of membrane receptors, whose function is usually modulated by G protein-coupled receptor kinases (GRKs). It has been reported that G protein-coupled receptors kinase 2 (GRK2) levels and activity are increased in patients and preclinical models of several relevant pathological conditions including cardiovascular disease, obesity, and insulin resistance [19]. In particular, GRK2 interacts with many signals involving transduction kinases (p38Mapk, AMPK, PI3K/Akt, MEK1), G protein subunits and modulators (Gαq, Gβγ, phosducin, RhoaA, RalA, EPAC1), and receptor tyrosine kinases and downstream molecules (EGFR, PDGFR, IRS1) [22].

In the VSMCs, agonists such as norepinephrine (NE) and PHE acting on GPCRs, specifically Gq/11, activate the phospholipase C (PLC) pathway generating inositol 1,4,5-trisphosphate (IP_3_) and diacylglycerol (DAG). IP_3_ binds to its receptors in the sarcoplasmic reticulum (SR), releasing Ca^2+^ into the cytosol; concomitantly, DAG stimulates protein kinase C (PKC), which phosphorylates various substrates including L-type VDCCs allowing Ca^2+^ entry into the cytosol. Other extracellular sources of Ca^2+^ include second messenger-operated Ca^2+^ channels and store-operated Ca^2+^ channels [23,24]. The increase in intracellular Ca^2+^ allows it to bind CaM, which leads to the activation of the myosin light-chain kinase (MLCK), which phosphorylates MLC20 producing actin-myosin coupling and vasoconstriction. Additionally, PKC phosphorylates and activates CPI-17, a smooth muscle-specific protein kinase inhibitor of MLC phosphatase, inhibiting MLCK and maintaining contraction [25]. Vasodilation of VSMCs is due to the decrease in cytosolic Ca^2+^, causing its dissociation from the CaM, MLCK inactivation, and MLCP-mediated MLC20 dephosphorylation, resulting in the dissociation of the actin-myosin interaction. Finally, Ca^2+^ output occurs via PMCA, and Ca^2+^ entry into SR occurs via the sarco/endoplasmic reticulum Ca^2+^-ATPase (SERCA) [26,27,28].

### 2.3. The Endothelial Control of VSMCs

Furchgott’s observation that the inner layer of ECs of the isolated aorta was essential to observe the relaxing response to ACh drastically changed the conception of the local control of vascular tone [1]. Subsequent studies showed that ECs cause arterial relaxation by releasing a vasoactive agent called endothelium-derived relaxing factor (EDRF)-identified later as NO-that stimulates the soluble guanylyl cyclase in VSMCs, increasing the production of cyclic guanosine monophosphate (cGMP) [29].

The synthesis of NO is performed by a family of enzymes called nitric oxide synthases (NOS) with three isoforms that exhibit different characteristics according to their functions in vivo [30]. There are two constitutive isoforms: endothelial NOS (or eNOS) that are primarily present in ECs, and neuronal NOS (or nNOS) that are primarily present in neurons. Both isoforms are Ca^2+^- and calmodulin-dependent [31]. The third isoform is an inducible type of NOS (or iNOS), independent of Ca^2+^ and calmodulin that is present in different cell types and expressed in abnormal cellular processes such as heart failure [32,33].

Other substances can also induce endothelium-dependent relaxation (EDR) including prostacyclin (PGI_2_) and endothelium-dependent hyperpolarizing factor (EDHF) [29]. Furthermore, ECs can also release an endothelium-derived contracting factor (EDCF) such as vasoconstrictor prostanoids, endothelin-1, uridine adenosine tetraphosphate, and NO itself [34]. One of the first steps in the sequence of events leading to cardiovascular disease is the reduction of endothelial cell capacity to release NO, phenomena that can be experimentally evidenced by an impairment of EDR. This alteration is known as endothelial dysfunction and is recognized as an indicator and predictor of cardiovascular disease [35,36].

## 3. Phytochemical Assays of Plant Extracts and Isolated Compounds

The phytochemical assays can be separated into the three types of substances evaluated: plant extracts, alkaloids, and non-alkaloids compounds. Table 1, Table 2 and Table 3 (at the end of Section 3.1, Section 3.2, and Section 3.3, respectively) summarize the data including the extract or compound tested, endothelium participation, the postulated mechanism of action, and the reference.

### 3.1. Plant Extracts

It has been more than 27 years since Renaud and de Lorgeril [37] suggested an inverse correlation between moderate wine consumption, especially red wine, and a lower risk of coronary heart disease. One of the main in vitro experimental milestones was performed in IRA by Fitzpatrick et al., [38] when they demonstrated that grape juices and grape skin extracts induced EDR in IRA. The effect was reproduced by polyphenols and their derivate molecules, quercetin and tannic acid, both compounds present in grape skins and the contraction induced by PHE was attenuated in IRA preincubated with grape skin extracts. The extracts also increased the cGMP levels in intact vascular tissue, and both relaxation and the increase in cGMP were abolished by competitive inhibitors of eNOS. Therefore, vasorelaxation induced by grape products seems to be mediated by the NO-cGMP pathway (Table 1) [38].

*Panax ginseng* C. A. Mey (*Araliaceae*), also called Asian or Korean ginseng, has long been traditionally used in Korea and China to treat several diseases. Ginsenosides are the main active compounds of *P. ginseng*. These have displayed a variety of bioactive effects including anti-oxidation, anti-proliferation, anti-inflammatory, antidiabetic, and vasodilation. Kim et al., [39] studied the potential relaxing effect of an extract of *P. ginseng*, which contained ginsenosides in IRA; the results showed that the extract with a high content of ginsenosides induced EDR, which was corroborated by ginsenoside Rg3 and ginsenoside Rg1, two major ginsenosides of the protopanaxatriol group. Ginsenoside Rg3 was about 100-fold more potent than ginsenoside Rg1. EDR induced by ginsenoside Rg3 was associated with the formation of cGMP. Assays using atropine, propranolol, nifedipine, N(G)-nitro-L-arginine (L-NNA), and methylene blue, among other pharmacological tools, showed that ginsenoside Rg3 was a major mediator of the EDR mediated by NO in response to ginsenosides in IRA, probably by the activation of tetraethylammonium (TEA)-sensitive K^+^ channels [39].

*Angelica sinensis* (Oliv.) Diels (*Apiaceae*), commonly known as Dong Quai or “female ginseng”, is a herb widely used in Chinese traditional medicine for its health benefits including cardiovascular conditions such as hypertension. Rhyu et al., [40] studied the vascular relaxation produced by the butanolic fraction of *R. angelica* and its possible mechanisms in IRA precontracted with norepinephrine (NE). The butanolic fraction reduced the CaCl_2_-induced vasoconstriction in a high-K^+^ depolarized medium to less than one-tenth the concentration needed to induce vasodilation. The aqueous fraction induced the transient acute relaxation, which was abolished by endothelial ablation and pretreatment with L-NNA. Ferulic acid, the main active component in *A. sinensis*, induced dilatation in IRA with and without endothelium [40].

*Panax notoginseng* (Burkill) F. H. Chen (*Araliaceae*) is a traditional Chinese herbal medicine containing *P. notoginseng* saponins as the main bioactive ingredient. These saponins are associated with many effects on the cardiovascular system including inhibition of platelet aggregation and the increase of coronary flow. It has been reported that the crude extract of *P. notoginseng* contains more than 50 saponins, and the five major compounds are the ginsenosides Rg1, Rb1, Re, and Rd, and notoginsenoside R1. Wang et al., [41] showed that the effects of *P. notoginseng* saponins and compounds of this extract induced IRA relaxation in an endothelium- and concentration-dependent manner with participation of the NO and COX pathways.

One of the most popular herbal medicines is *Ginkgo biloba* L. (*Ginkgoaceae*), which has a long story of use including the clinical one to improve peripheral vascular diseases. Kubota et al., [42] found that the extract of *G. biloba* induced a dose-dependent relaxation in IRA precontracted with NE, which was abolished by N(ω)-nitro-L-arginine methyl ester (L-NAME). In addition, quercetin, one component of the extract, produced a similar relaxation that was also suppressed by L-NAME. The authors, with complementary studies measuring cytosolic Ca^2+^ in cultured aortic ECs, suggested that the effect induced by the *G. biloba* extract is mainly due to quercetin activating NO synthesis and release by increasing Ca^2+^ in vascular ECs [42].

Organic extracts enriched in polyphenols prepared from palm fronds (*Elaesis guineensis* Jacq.), lemongrass (*Cymbopogon citrates* (DC.) Stapf), papaya shoots (*Carica papaya* L.), and green chili (*Capsicum frutescens* L.) were tested for their ability to induce IRA relaxation. The palm frond extract produced an EDR relaxation (>75%) similar to ACh and was significantly greater than that produced by the lemongrass extract, papaya shoots, and green chili; the last three extracts achieved an average relaxation between 35% and 45% [43].

*Arbutus unedo* L. (*Ericaceae*) is an evergreen bush of the Mediterranean basin and regions with hot summers and a mild rainy winter. This plant has been used in oriental Morocco to treat arterial hypertension. Assays conducted by Ziyyat et al., [44] with the aqueous extract of *A. unedo* roots (0.25 mg/mL) showed an EDR of IRA precontracted with NE; moreover, the relaxation was inhibited by the pretreatment of vessels with L-NMA, methylene blue, or 1H-[1,2,4] oxadiazolo [4,3-a] quinoxaline-1-one (ODQ; a guanylate cyclase inhibitor), but not by atropine. These results suggest that the extract produced an EDR of IRA primarily mediated by a stimulation of eNOS by mechanisms other than the activation of muscarinic receptors [44].

*Eucommia ulmoides* Oliv. (*Eucommiaceae*) has been a common active ingredient in traditional herbal prescriptions in China for hypertension. To probe the potential vasodilatory effect of this plant, aqueous extracts from *E. ulmoides* leaf and bark were prepared and tested in IRA [45]. Aqueous extracts obtained from the leaf and bark of *E. ulmoides* induced a concentration-dependent and EDR in IRA precontracted with PHE. Both IRA precontracted with high K^+^ and that without endothelium precontracted with PHE did not respond to the extracts. The EDR induced by the extracts was abolished by L-NAME and was markedly reduced by methylene blue, denoting the participation of the NOS pathway in the observed vasorelaxation. The relaxation induced by the extracts was not inhibited by atropine, but was inhibited by TEA and 4-aminopyridine, suggesting that the EDR was not mediated via the activation of endothelium muscarinic receptors and may therefore involve the activation of K^+^ channels. According to the authors, this study provides in vitro evidence of the use of *E. ulmoides* as an antihypertensive [45].

*Schizandra chinensis* (Turcz.) Baill. (*Schisandraceae*) is one plant species that is part of the Chinese medicinal formula ‘Shengmai San’, which is commonly used to treat coronary heart disease. It has been demonstrated that the extract of *S. chinensis* fruit activates estrogen receptors [46]. There is accumulating evidence that phytoestrogens exert favorable effects on cardiovascular disease by directly modulating arterial tone via multiple cellular mechanisms. On this basis, the capability of an aqueous extract of *S. chinensis* fruit to induce relaxation in IRA precontracted with NE was tested. The extract induced partial EDR, where relaxation induced by lower concentrations (0.1 and 0.3 mg/mL) was largely endothelium-dependent and abolished by L-NNA, methylene blue, ODQ, indomethacin, or ICI 182,780. According to the authors, these results indicate that relaxation induced by the extract is associated with the NO-cGMP pathway via estrogen receptors. Since the magnitude of the inhibition with these treatments decreased with high extract concentrations, it has been proposed that other mechanisms of relaxation would be involved [47].

*Cirsium japonicum* (Thunb.) Fisch. ex DC. (*Compositae*) is a herb located in many areas of Korea, China, and Japan, and has been used in Chinese medicine to treat hypertension among other illness. Kim et al., [48] evaluated the potential vasodilatory activity of the aqueous extract of *C. japonicum* and its possible mechanism in IRA precontracted with NE. The extract induced relaxation in IRA with intact endothelium was in a concentration-dependent manner (0.1–1.0 mg/mL). The relaxation was abolished by the removal of the endothelium and pretreatment with L-NNA, methylene blue, or diphenylhydramine; indomethacin, atropine, and substance P did not affect the relaxation. In agreement with the results, the authors indicate that the response to the *C. japonicum* extract involved an enhancement of the NO-cGMP pathway via histamine H1-receptors [48].

Based on previous reports indicating that procyanidins induced a decrease of blood pressure, Matsui et al., [49] conducted a study evaluating the effect of procyanidins in IRA precontracted with PHE. Procyanidins (0.3–100 µg/mL) obtained from apple (*Malus domestica* Mill.) induced an EDR of the IRA. The effect was entirely abolished by ODQ, revealing that apple procyanidins were associated with the cGMP production pathways. Thus, the authors suggested that vasodilatation induced by apple procyanidins was related to the NO-cGMP pathway [49].

Another plant that has been used to treat cardiovascular disease for over a millennium in eastern countries and is primarily found in China, Japan, and Korea is *Morus alba* L. (*Moraceae*). Xia et al., [50] assayed the vasoactive effect and the mechanism associated with an ethyl acetate extract from leaves of *M. alba* in IRA. The extract (0.125–32.000 g/L) induced a concentration-dependent relaxation in IRA both with and without endothelium precontracted by high K^+^ or PHE. In endothelium-free IRA, the extract at the EC_50_ reduced Ca^2+^-induced contraction after PHE- or K^+^-generated stable contraction in the Ca^2+^-free solution. After incubation with verapamil, the extract induced contraction in endothelium-free IRA precontracted by PHE, an effect that was abolished by ruthenium red, but not by heparin. The results showed that the extract of *M. alba* had dual vasoactive effects, and that the relaxation was higher than the contraction. Therefore, the authors proposed that relaxation was mediated by inhibition of voltage- and receptor-dependent Ca^2+^ channels in VSMCs, whereas contraction was associated with the activation of ryanodine receptors in the sarcoplasmic reticulum [50]. Recently, Panth et al., [51] investigated the effects of the ethanol extract of root bark obtained from *M. alba* in IRA and found that the extract induced concentration-dependent vasorelaxation following an endothelium-dependent pathway. Accordingly, it was proposed that the root bark extract of *M. alba* produces a potent EDR involving the NO-cGMP pathway in combination with K^+^ channel activation [51].

*Curcuma comosa* Roxb. (*Zingiberaceae*) is a plant native to Asia that is used in traditional Thai medicine as an anti-inflammatory agent for the treatment of postpartum uterine bleeding. Since *C. comosa* contains phytoestrogens, Intapad et al., [52] examined its vasoactive effect in IRAR. The acute exposure of IRA with and without endothelium to the hexane extract of *C. comosa* or an isolated diarylheptanoid compound, D3, did not induce relaxation. However, preincubation of aortic rings for 20 min with the extract or the isolated diarylheptanoid compound enhanced EDR induced by ACh. The hexane extract did not modulate the relaxation in endothelium-free IRA in response to sodium nitroprusside (SNP), suggesting a predominant effect on ECs rather than on VSMCs. Coincubation with ICI 182,780 (an estrogen receptor antagonist), L-NAME, or ODQ inhibited the enhancing effects of *C. comosa* on ACh-induced relaxation. The authors suggested that the effect of *C. comosa* was mediated through the estrogen receptor and NO-cGMP-dependent mechanisms; moreover, *C. comosa* also increased the phosphorylation of serine 1177 eNOS and serine 473 Akt proteins, and these effects were abolished by ICI 182,780. It has been suggested that *C. comosa* acutely increases the EDR of IRA through the ER-Akt-eNOS pathway, suggesting a non-genomic action of a phytoestrogen from *C. comosa* on vascular relaxation [52].

*Xanthoceras sorbifolia* Bunge (*Sapindaceae*) is a shrub distributed in Inner Mongolia, China, and has been used in traditional medicine to treat various illnesses including hypertension. Jin et al., [53] aimed to investigate the potential vasodilatory effect of the ethanol extract of *X. sorbifolia* on PHE-precontracted IRA tension and its associated mechanisms. The extract (0.1–100 µg/mL) induced relaxation on the PHE-precontracted IRA in a concentration-dependent manner. The effect was endothelium-dependent since its elimination abolished the vasorelaxation; the same effect also had pretreatment of intact vessels with L-NAME and ODQ [44]. The vasodilation was not affected by the blockade of L-type VDCCs; however, depletion of extracellular Ca^2+^, thapsigargin (a modulator of the store-operated Ca^2+^ entry, SOCE), and wortmannin (an inhibitor of Akt) significantly attenuated the vasodilation induced by the extract of *X. sorbifolia*. Consequently, it has been proposed that the extract relaxes VSMCs via endothelium-dependent NO-cGMP signaling through activation of the Akt- and SOCE-eNOS-sGC pathways, which may be associated with the function of K^+^ channels [53].

*Combretum leprosum* Mart. (*Combretaceae*) is a shrub that grows in northeastern Brazil and the extracts from different parts are used to treat many illnesses. Alves Filho et al., [54] assayed the ethanolic extract of the bark of *C. leprosum* in IRA. The extract (1.5 µg/mL) induced EDR that was significantly reduced in the absence of extracellular Ca^2+^, and the relaxation was reverted by ruthenium red (a transient receptor potential channels blocker). Thus, it is suggested that the *C. leprosum* extract contains bioactive compounds that induce Ca^2+^-dependent stimulation of ECs, allowing a prolonged production of relaxing factors [54].

*Calpurnia aurea* subs. *aurea* (Aiton) Benth. (*Leguminosae*) is a shrub widely distributed in Africa and used in traditional medicine to treat some illnesses including hypertension. On this basis, Getiye et al., [46,55] performed a study to evaluate the antihypertensive activity of the methanol extract of *C. aurea* seeds. The extract caused a dose-dependent relaxation of IRA precontracted with high K^+^, and the relaxation mechanism was found to be independent of muscarinic and histamine receptors, ATP-dependent K^+^ channels, cyclooxygenase enzymes, the -NO pathway, and the endothelium. On this basis, it is suggested the participation of VDCCs in *C. aurea* induces IRA relaxation [55].

*Heliopsis longipes* (A. Gray) S. F. Blake (*Compositae*) is a herbaceous plant native to Mexico where the roots of this species are used as a spice, home insecticide, and for the treatment of some illnesses including toothaches and muscular pain. Castro-Ruiz et al., [56] studied the effect of the ethanolic extract of *H. longipes* roots in IRA precontracted with PHE. They found that the extract produced an EDR in IRA and proposed that this effect was mediated by gasotransmitters and prostacyclin signaling pathways [56].

*Chenopodium ambrosioides* L. (*Chenopodiaceae*) is an annual or short-lived perennial herb native to Central America, South America, and southern Mexico that is traditionally used as an anthelmintic and to treat many other symptoms including fever, influenza, migraine, and hypertension. Assaidi et al., [57] prepared methanolic extracts from the aerial part of *C. ambrosioides* and evaluated their vasodilatory potential in IRA. The ethanolic extract of *C. ambrosioides* induced a concentration- and endothelium-dependent dilatation of IRA precontracted by PHE. Furthermore, the effect of the extract was inhibited by atropine and TEA (a non-selective K^+^ channel blocker), while the effect remained unchanged in the presence of L-NAME and glibenclamide. From the results, the authors concluded that the ethanolic extract of the aerial part of *C. ambrosioides* induced EDR in IRA through the stimulation of muscarinic receptors present in ECs and probably activated the Ca^2+^-activated K^+^ channels [57].

In our laboratory, we studied the vasodilatory effect of *Centaurium cachanlahuen* (Mol.) Robinson (*Gentianiaceae*) [58] and *Solanum crispum* Ruiz & Pav. (*Solanaceae*) [59]. *C. cachanlahuen* is a Chilean native plant widely used in traditional medicine for its antirheumatic, analgesic, anti-inflammatory, and hypotensive therapeutic effects. We evaluated the relaxing effect of the aqueous and hydroalcoholic extracts of *C. cachanlahuen* in IRA precontracted with PHE. The results showed that both the aqueous (3 mg/mL) and the hydroalcoholic extracts (3 mg/mL) induced IRA vasodilatation higher with the hydroalcoholic extract when compared to the aqueous extract. This effect was highly endothelium-dependent since it was almost abolished in endothelium-free IRA such as in vessels with endothelium preincubated with L-NNA, indicating the participation of NOS in the relaxation induced by the extracts of *C. cachanlahuen*. Considering the high relative concentration of xanthones in the extracts, we suggest that the compound may participate in the vasodilator effect of the extracts of *C. cachanlahuen*. Therefore, we provided experimental in vitro evidence that supported the folk use of *C. cachanlahuen* as an infusion to treat hypertension [58].

*S. crispum* is a southern South American native plant frequently used in Chilean folk medicine including the Mapuche communities from northwestern Patagonia to treat a wide variety of symptoms associated with both acute and chronic ailments. It has been reported to have analgesic, anti-inflammatory, and antihypertensive properties. The assays carried out in our laboratory showed that the hydroalcoholic extract of *S. crispum* induced relaxation of IRA precontracted with PHE that reached mean values of 43%. The relaxation was largely endothelium-dependent and mediated by NO, since it was significantly reduced in vessels without endothelium as in vessels preincubated with L-NNA. Similar to *C. cachanlahuen*, this result also supports the popular use of *S. crispum* in the treatment of hypertension [59].

*Crocus sativus* L. (*Iridaceae*), commonly known as saffron, is a perennial herb widely cultivated in Spain, Iran, and other countries. Saffron is extensively used as a food additive and is also used in traditional and alternative medicine to treat many illnesses. Razavi et al., [60] assayed the vasodilatory effects of an aqueous saffron extract (0.5, 1, and 2 mg/mL) in IRA precontracted with PHE or high K^+^. Saffron induced EDR in IRA precontracted with PHE in a concentration-dependent manner. The relaxation induced by the highest saffron concentration in vessels with endothelium precontracted with high K^+^ was less than that observed in vessels precontracted with PHE. The vasodilatory effect of saffron was abolished by L-NAME and partially abolished in endothelium-free IRA. From the results, the authors proposed that relaxation induced by saffron is mainly related to the NOS pathway and partly related to L-type VDCCs [60].

### 3.2. Alkaloids

Alkaloids are organic compounds with nitrogen in their heterocyclic ring that are present in plants, marine organisms, fungi, and microorganisms, however, alkaloids also include metabolites with nitro groups in the form of amide groups, nitrogen atoms in the side chain, or in other functional groups. Alkaloids can be found as water-soluble salts of organic acids, esters, combined with tannins or sugars or as free bases [61].

The first assay showed in this section is the effect of tetrandrine, a bisbenzylisoquinoline alkaloid isolated from *Stephania tetrandra* S. Moore (*Menispermaceae*), which is traditionally used in China for treating inflammation, hypertension, and silicosis; tetrandrine shows a potent antihypertensive effect that has been demonstrated in many animal species. In IRA, tetrandrine blocked VDCCs in VSMCs with higher sensitivity than those presented in isolated rat atria (Table 2) [62].

Berberine is a benzylisoquinoline alkaloid extracted from the rhizome of Chinese herbs such as *Coptis chinensis* Franch. (*Ranunculaceae*), *Coptis deltoidea* C. Y. Cheng & P. K. Hsiao (*Ranunculaceae*), and *Coptis teeta* Wall (*Ranunculaceae*). The study focused on the effect of the low concentration of berberine since, according to the authors, it is more representative of the therapeutic concentration in Chinese herbal medicine. Thus, berberine (1 × 10^−6^ M) induced IRA relaxation that was endothelium-dependent, while higher concentrations produced both EDR and endothelium-independent relaxation (EIR). Based on this experimental evidence, it is suggested that the EDR induced by berberine in IRA participates the release of NO [63].

Thaligrisine is a bisbenzyl tetrahydroisoquinoline alkaloid isolated from the bark of *Pseudoxandra esclerocarpa* (*Annonaceae*) and structurally related to tetrandrine. This alkaloid inhibits IRA vasoconstriction induced by NE and high K^+^. Tur et al., [64] showed evidence that thaligrisine has a higher affinity for [^3^H]-prazosin binding site than [^3^H]-(+)-*cis*-diltiazem binding sites, which was different from the result observed for tetrandrine and isotetrandrine that presented a similar affinity for both receptors [64].

Mesaconitine is a diterpene alkaloid and is one of the aconite alkaloids present in *Aconitum* Thunb. (*Ranunculaceae*), an oriental herb ancestrally used in Japan and China to improve physical conditions. According to the investigators, mesaconitine inhibited the IRA contraction induced by PHE only in vessels with an intact endothelium, an effect that was dependent on external Ca^2+^ concentration [65].

More recently, Martínez et al., [66] designed a study to explore the effect of antioquine in IRA precontracted with high K^+^. Antioquine is a bisbenzylisoquinoline alkaloid isolated from *Pseudoxandra esclerocarpa* Maas. (*Annonaceae*), a Colombian endemic plant from the region of Antioquia [67]. They compared antioquine with tetrandrine and verapamil. Results showed that verapamil blocked IRA contraction in vessels with and without endothelium, while antioquine and tetrandrine induced a blockade near 50% without significant differences between the vessels with and without endothelium [66]. Based on these results, it can be assumed that antioquine, like other isoquinoline alkaloids, induced vasodilation by blocking VDCCs in VSMCs.

*Aristotelia chilensis* ([Molina], Stuntz) a member of the family *Eleocarpaceae*, is a plant native to Chile distributed in Asia, Australia, the Pacific Area, and South America. *A. chilensis* produces several compounds with bioactive potential including flavonoids in fruits. Cifuentes et al., [68] isolated and purified 8-oxo-9-dihydromakomakine, an alkaloid and the main component of *A. chilensis* leaf. They assayed 8-oxo-9-dihydromakomakine activity in IRA precontracted with PHE. Results showed that 8-oxo-9-dihydromakomakine induced endothelium-independent IRA relaxation; the alkaloid especially reduced the IRA contraction induced by high K^+^ and those induced by Bay K-6844 (an agonist of L-type VDCCs). The researcher proposed that 8-oxo-9-dihydromakomakine could probably decrease Ca^2+^ entry through VDCCs [68].

### 3.3. Non-Alkaloid Compounds

Polyphenols are an extensive class of natural molecules with many biological activities [69]. Resveratrol (3,4’,5-trihydroxystilbene) and quercetin (3,3’,4,5,7-pentahydroxyflavone) are two representatives of this class of compounds that have been extensively studied. In 1996, Chen and Pace-Asciak [70] aimed to test whether resveratrol and quercetin were capable of producing vasodilation in IRA. Results showed that both resveratrol and quercetin dose-dependently inhibited the contractile response to norepinephrine, but that effect was abolished by pretreatment of the vessel with L-NNA. Additionally, resveratrol and quercetin induced the relaxation of IRA precontracted with PHE, an effect that was also abolished by L-NNA. Interestingly, high concentrations of resveratrol and quercetin also relaxed the IRA without endothelium. From these results, it was concluded that resveratrol and quercetin induce and exert both indirect and direct vasodilator effects on the IRA by NOS-mediated and non-NOS-mediated pathways, respectively (Table 3) [70].

Chan et al., [71] evaluated the relaxing effect and the structure-activity relation of three flavonols (fisetin, quercetin, and 3,3’,4’-trihydroxyflavone) and three flavones (apigenin, chrysin, and luteolin) in IRA; the control responses to these flavonoids were compared with the responses obtained in vessels without endothelium or in the presence of nifedipine (10^−7^ M, an L-type VDCCs blocker). They also investigated the effects of flavonoids on the contraction induced by extracellular Ca^2+^ and on the release of intracellular Ca^2+^ induced by agonists. Flavones showed an endothelium-independent relaxation, while flavonols showed an endothelium-dependent component. Endothelium removal decreased the sensitivity of responses to flavonols without affecting the maximal relaxation. In the presence of nifedipine, relaxation induced by apigenin, luteolin, and quercetin was inhibited, but relaxation induced by chrysin, fisetin, and 3,3’,4’-trihydroxyflavone was not affected. All compounds assayed showed a concentration-dependent inhibition of the contractile responses induced by Ca^2+^ and the release of intracellular Ca^2+^ stimulated by PHE [71].

Xanthorrhizol is a sesquiterpene isolated from *Iostephane heterophylla* (Cav.) Benth. (*Compositae*), a medicinal plant used in Mexico to treat wounds and sores, rheumatism, arthritis, diabetes, gastrointestinal complaints, and liver ailments. Campos et al., [72] assayed the effect of xanthorrhizol in IRA and found that this compound (1, 3, 10, 30, and 100 µg/mL) induced the relaxation of vessels precontracted by high K^+^ (60 mM), NE (10^−6^ M), or CaCl_2_ (1.0 mM). The effect of xanthorrhizol was not affected by indomethacin (10 µM) or L-NAME (100 µM), suggesting that the effect was not mediated by either PGI_2_ or NO released from ECs. The inhibitory effect of xanthorrhizol was greater on K^+^- and Ca^2+^-induced contractions than those induced by NE. On this basis, it was suggested that the inhibitory effect of xanthorrhizol may explain the interference with Ca^2+^ availability by inhibiting Ca^2+^ entry through both the voltage and receptor-gated Ca^2+^ channels [72].

The same researchers studied *Vernonia* spp. (*Asteraceae*), a plant used in Latin America traditional medicine to treat menstrual and stomach disorders, suggesting smooth muscle relaxing properties of some of their chemical constituents. To support these uses, Campos et al., [73] studied the potential vasorelaxant effect of the sesquiterpene lactones, glaucolides D and E, in IRA. The concentration–response curve showed that glaucolide E was more potent than glaucolide D to relax IRA precontracted with high K^+^ or NE. The authors suggest that the relaxant effect induced by the glaucolides indicate that they act by the inhibition or blockade of Ca^2+^ entry through VDCCs- and receptor-operated Ca^2+^ channels where glaucolide E has a higher affinity than glaucolide D for Ca^2+^ channels [73].

Chan et al., [74] studied *Ligusticum chuanxiong* Hort. (*Apiaceae*), a commonly used traditional Chinese medicinal herb to treat cardiovascular diseases in China. They investigated the vasorelaxation effects of ligustilide and senkyunolide A, two phthalide derivatives and the most abundant constituents of the plant. Both ligustilide and senkyunolide A showed similar relaxation potencies in IRA precontracted with U46619 (a prostaglandin derivate), PHE, 5-hydroxytryptamine, and high K^+^. Their relaxation effects were not affected by endothelium removal or by the presence of an adenylate cyclase inhibitor, sGC inhibitor, or TEA. Regarding the underlying mechanism of action, the authors indicated that further investigations are required [74].

Cornuside is a bisiridoid glucoside compound isolated from the fruit of *Cornus officinalis* Siebold & Zucc. (*Cornaceae*); *C. officinalis* is a commonly used traditional Chinese medicine and its fruits have long been used to treat kidney deficiency, high blood pressure, and other diseases in China. Therefore, Kang et al., [75] investigated the effects of cornuside on the IRA to account for its beneficial effect on the cardiovascular system. Cornuside induced a concentration-dependent relaxation of IRA precontracted with PHE that was abolished with endothelial removal. Pretreatment of IRA with either L-NAME or ODQ completely inhibited the relaxation induced by cornuside. The effect of cornuside was not blocked by pretreatment with verapamil, diltiazem, TEA, glibenclamide (a selective ATP-sensitive K^+^ channel blocker), indomethacin (a non-specific cyclo-oxygenase inhibitor), atropine (a non-selective muscarinic antagonist), or propranolol (a non-selective beta-blocker), suggesting that cornuside induces EDR via endothelium-dependent NO/cGMP signaling [75].

The fruit of *S. chinensis* is known to be enriched in lignans, among them is gomisin A. Park et al., [76] studied its vasodilatory effect in IRA and found that gomisin A (1 × 10^−6^ to 3 × 10^−4^ M) induced a concentration-dependent relaxation that was markedly decreased by endothelium removal and by pretreatment with L-NNA or ODQ. In IRA without endothelium, the small relaxation induced by gomisin A by GA was inhibited by calyculin A (a phosphatase inhibitor). Based on these results, the authors suggest that gomisin A induced vascular relaxation by partially activating EDR and partial dephosphorylation of the myosin light chain [76].

Scutellarin is a flavonoid extracted from the traditional Chinese herb, *Erigeron breviscapus* Hand.–Mazz. (*Compositae*), which has been used in China for a long time to relieve cardiovascular diseases including hypertension. Pan et al., [77] carried out a study to test the potential vasorelaxant effect of scutellarin in IRA and found that 3, 10, 30, and 100 µM induced a dose-dependent relaxation in both endothelium-intact and endothelium-free IRA precontracted with NE, but not with high K^+^. TEA, glibenclamide, atropine, propranolol, indomethacin, and L-NAME did not affect the scutellarin response and excluded K^+^ channels, the muscarinic receptor, NO pathway, and prostaglandin in its effect. When IRA was pretreated with scutellarin and then contracted with NE, the tonic phase decreased without affecting the phasic phase. Scutellarin also inhibited the increase in cytosolic Ca^2+^ induced by NE. It has been proposed that scutellarin seems to relax IRA in an endothelium-independent manner and that the mechanism could be associated with the inhibition of extracellular Ca^2+^ entry independent of VDCCs [77].

Lectins belong to a protein group that can bind reversibly to specific carbohydrates present in many organisms. Plant lectins have been associated with many roles including the regulation of mediators in vessels. On this basis, Assreuy et al., [78] evaluated the vasorelaxant effect of the seed lectin of *Pisum arvense* L. (*Leguminosae*) in IRA with and without endothelium, precontracted with PHE. The lectin induced a concentration-dependent relaxation in IRA that was reversed by L-NAME and glucose; the relaxation was totally inhibited in Ca^2+^-free buffer, but not in the presence of atropine. The authors suggest that there is interaction between the lectin carbohydrate-binding sites with specific receptors located in the vascular ECs, allowing the activation of NOS [78].

Farrerol is a typical flavanone isolated from the Chinese medicinal plant *Rhododendron dauricum* L. (*Ericaceae*). It has been widely used to mitigate symptoms related to bronchial asthma in China. The evidence also suggests that farrerol possesses many biological properties including antibacterial, anti-inflammatory, and inhibitory effects on VSMC proliferation. Qin et al., [79] studied the vasoactive effect of farrerol in IRA and found that farrerol induced a dose-dependent and endothelium-independent relaxation. In endothelium-free IRA and in the presence of a stable contraction induced by high K^+^ or PHE in Ca^2+^-free solution, farrerol reduced the contraction induced by Ca^2+^. From these results, it was suggested that farrerol may block the Ca^2+^ entry via L-type VDCCs [79].

In our laboratory, we tested the vasodilatory potential of *p*-coumaric acid (3-[4-hydroxyphenyl]-2-propenoic acid), a hydroxyl derivative of cinnamic acid widely distributed in fruits [80]. *p*-coumaric acid showed a moderate endothelium-dependent vasodilatory effect in IRA, reaching about 30% relaxation with respect to the maximal relaxation induced by ACh. Additionally, *p*-coumaric acid protects the relaxation produced by ACh when it is coincubated with IRA exposed to high glucose, a condition that significantly deteriorated the ACh-induced vasodilation [80].

Brazilin is a natural heterotetracyclic compound and a red pigment obtained from the wood of *Caesalpinia echinata* Lam (*Leguminosae*). This compound shows many biological activities including antidiabetic, anti-inflammatory, anti-platelet aggregation, and anti-oxidation. Yan et al., [81] studied its potential vasoactive effect in IRA and discovered that brazilin (10–100 µM) induced a dose-dependent relaxation of vessels precontracted with NE or high K^+^. The effect induced by brazilin was attenuated by endothelium removal or preincubation with L-NAME, methylene blue, or indomethacin. Based on these results, it was proposed that brazilin induces relaxation in IRA via both endothelium-dependent and -independent ways, and attenuates vasoconstriction via blocking VDCCs and receptor-operated Ca^2+^ channels [81].

It is well-known that caffeine is the most widely consumed psychoactive substance in the world. Experimental data obtained from vessels show that caffeine produces both vasoconstriction and vasodilation, the first effect activating ryanodine receptors and releasing Ca^2+^ from intracellular stores and the second one mediated by an increase of NO in ECs or an inhibition of phosphodiesterase in VSMCs. Latif and Badar [82] set out to investigate the effect of caffeine in IRA and the role of K^+^ channels. IRA in a state of contraction or relaxation were exposed to caffeine and various K^+^ channel blockers; under these conditions, caffeine-induced IRA relaxation occurred in both the absence or presence of NE, a response that was blocked in vessels pretreated with K^+^ channel blockers. The investigators proposed that the vasodilatory effect of caffeine may involve ATP-dependent, Ca^2+^-mediated, or voltage-dependent K^+^ channels [82].

## 4. Conclusions

Through the years, the IRA model has been a useful tool to validate vasoactive properties of various substances, particularly the vasodilator effect of extracts and compounds obtained from plants used in traditional medicine to treat hypertension, among other ailments. The relaxation induced by these extracts largely depends on the presence of a functional endothelium and usually involves the NOS pathway.

Many of the revised vasoactive alkaloids share an isoquinoline structure, except for mesaconitine and 8-oxo-9-dihydromakomakine. Those isoquinoline alkaloids show the ability to induce vasorelaxation in an endothelium-independent manner by blocking Ca^2+^ entry into the VSMCs and producing IRA relaxation. The mechanism of action of isoquinoline alkaloids seems to be related to the L-type VDCCs-blocking activity in VSMCs. This action is in agreement with the mechanism described for tetrandrine.

Among the non-alkaloid compounds, we also highlighted polyphenols, particularly flavonoids that seem to be associated with endothelium-dependent IRA relaxation. Other compounds like bisiridoid glucoside and lectins also induce EDR, while the relaxation induced by sesquiterpenes and phthalide derivatives are endothelium-independent. The lignan gomisin A seems to produce both endothelium-dependent and -independent vasodilatation.

## Figures and Tables

**Figure 1 animals-09-00623-f001:**
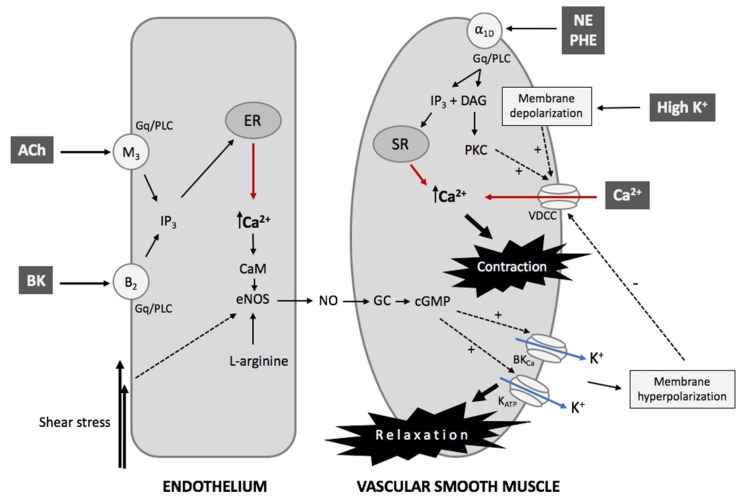
Interaction between endothelium and vascular smooth muscle cells in the aorta. The figure is a simplified representation showing the main component determining aortic contractibility associated with the isolated rat aorta (IRA) model used to evaluate the bioactivity of phytochemicals. ER: endoplasmic reticulum; SR: sarcoplasmic reticulum; ACh: acetylcholine; BK: bradykinin; PHE: phenylephrine; NE: norepinephrine; M_3_: muscarinic type-3 acetylcholine receptor; B_2_: bradykinin type-2 receptor; α_1D_: alpha-1D-adrenergic receptor; Gq/PLC: Gq-protein/phospholipase C; IP_3_: inositol 1,4,5-trisphosphate; DAG: diacylglycerol; PKC: protein kinase C; CaM: calmodulin; eNOS: endothelial nitric oxide synthase; NO: nitric oxide; GC: guanylate cyclase; cGMP: cyclic guanosine monophosphate; BKca: large-conductance Ca^2+^-activated K^+^ channel; K_ATP_: ATP-sensitive K^+^ channel; VDCC: voltage-dependent Ca^2+^ channel.

**Table 1 animals-09-00623-t001:** Summary of vasodilator effects induced by plant extracts in isolated rat aorta (IRA).

Plant Extracts	Endothelium Participation	Postulated Mechanism	Reference
*Vitis vinifera* (grape products)	EDR	NOS/cGMP pathway	[38]
*Panax ginseng* (ginsenosides)	EDR	NOS/TEA-sensitive K^+^ channels	[39]
*Radix angelica* (aqueous and butanol fraction)	EDR and EIR	NOS pathway	[40]
*Panax notoginseng* (saponins)	EDR	NOS and COX pathways	[41]
*Ginkgo biloba*	EDR	NOS pathway	[42]
*E. guineensis*, *C. citrates*, *C. papaya*, and *C. frutescens* (Extracts enriched in polyphenols)	EDR		[43]
*Arbutus unedo* (aqueous extract)	EDR	NOS pathway	[44]
*Eucommia ulmoides* (aqueous extracts)	EDR	NOS pathway	[45]
*Schizandra chinensis* (aqueous extract)	EDR	NOS pathway/via estrogen receptors	[47]
*Cirsium japonicum* (aqueous extract)	EDR	NOS/cGMP pathway/via histamine H_1_-receptors	[48]
*Malus pumila* (apple procyanidins)	EDR	NOS/cGMP pathway	[49]
*Morus alba* (ethyl acetate extract from leaves)	EIR	Inhibition of voltage- and receptor-dependent Ca^2+^ channels in VSMCs	[50]
*Morus alba* (ethanol extract from root bark)	EDR	NOS pathway/K^+^ channel activation	[51]
*Curcuma comosa* (hexane extract)	Increase ACh-induced EDR	Estrogen receptor/NO-cGMP-dependent mechanisms and ER-Akt-eNOS pathway	[52]
*Xanthoceras sorbifolia* (ethanol extract)	EDR	NOS-cGMP pathways and Akt- and SOCE-eNOS-sGC pathways	[53]
*Combretum leprosum* (ethanol extract)	EDR	Ca^2+^-dependent stimulation of ECs	[54]
*Calpurnia aurea* (methanol extract)	EIR	It is suggested VDCCs participation.	[55]
*Heliopsis longipes* (ethanolic extract)	EDR	It is suggested gasotransmitters and prostacyclin participation.	[56]
*Chenopodium ambrosioides* (methanolic extracts)	EDR	Stimulation of muscarinic receptors in ECs	[57]
*Centaurium cachanlahuen* (aqueous and hydroalcoholic extracts)	EDR	NOS pathway	[58]
*Solanum crispum* (hydroalcoholic extract)	EDR	NOS pathway	[59]
*Crocus sativus* (aqueous extract)	EDR and EIR	NOS pathway and VDCCs in VSMCs	[60]

**Abbreviations:** ECs: endothelial cells; VSMCs: vascular smooth muscle cells; EDR: endothelium-dependent relaxation; EIR: endothelium-independent relaxation; NOS: nitric oxide synthase; eNOS: endothelial nitric oxide synthase; cGMP: cyclic guanosine monophosphate; VDCCs: voltage-dependent Ca^2+^ channels; TEA: tetraethylammonium; COX: cyclooxygenase; SOCE: store-operated Ca^2+^ entry; sGC: soluble guanylate cyclase.

**Table 2 animals-09-00623-t002:** Summary of vasodilator effects induced by alkaloids in isolated rat aorta (IRA).

Alkaloids	Endothelium Participation	Postulated Mechanism	Reference
Tetrandrine (bisbenzylisoquinoline alkaloid)	EIR	Blocked VDCCs in VSMCs	[62]
Berberine (benzylisoquinoline alkaloid)	EDR/EIR	EDR (low concentrations) EIR (high concentrations)	[63]
Thaligrisine (bisbenzyl tetrahydroisoquinoline alkaloid)	EIR	Blocked VDCCs in VSMCs	[64]
Mesaconitine (diterpene alkaloid)	EIR	Blocked VDCCs in VSMCs	[65]
Antioquine (bisbenzylisoquinoline alkaloids)	EIR	Blocked VDCCs in VSMCs	[66]
8-Oxo-9-dihydromakomakine (isolated from *Aristotelia chilensis*)	EIR	Blocked VDCCs in VSMCs	[68]

**Abbreviations:** VSMCs: vascular smooth muscle cells; EDR: endothelium-dependent relaxation; EIR: endothelium-independent relaxation; VDCCs: voltage-dependent Ca^2+^ channels.

**Table 3 animals-09-00623-t003:** Summary of vasodilator effects induced by alkaloids in isolated rat aorta (IRA).

Non-Alkaloid Compounds	Endothelium Participation	Postulated Mechanism	Reference
Resveratrol and quercetin (polyphenols)	EDR and EIR	NOS (low concentrations) and non-NOS (high concentrations) pathways.	[70]
Flavonols (fisetin, quercetin, and 3,3’,4’-trihydroxyflavone) and flavones (apigenin, chrysin, and luteolin)	EDR and EIR	NOS (flavonols) and non-NOS (flavones) pathways.	[71]
Xanthorrhizol (sesquiterpene)	EIR	Inhibiting voltage/receptor-gated Ca^2+^ channels.	[72]
Glaucolides D and E	EIR	Inhibiting voltage/receptor-gated Ca^2+^ channels.	[73]
Ligustilide and senkyunolide A (phthalide derivatives)	EIR		[74]
Cornuside (bisiridoid glucoside)	EDR	NOS/cGMP pathways	[75]
Gomisin A (lignan)	EDR	NOS pathway.	[76]
Scutellarin (flavonoid)	EIR	Blocked Ca^2+^ entry independent of VDCCs.	[77]
Lectin (seed lectin of *P. arvense*)	EDR	NOS pathway.	[78]
Farrerol (flavanone)	EIR	Blocked VDCCs in VSMCs	[79]
*p*-Coumaric acid	EDR	NOS pathway.	[80]
Brazilin (heterotetracyclic)	EDR and EIR	Blocked VDCCs in VSMCs	[81]
Caffeine	EDR and EIR	Voltage-dependent K^+^ channels.	[82]

**Abbreviations:** VSMCs: vascular smooth muscle cells; EDR: endothelium-dependent relaxation; EIR: endothelium-independent relaxation; NOS: nitric oxide synthase; synthase; cGMP: cyclic guanosine monophosphate; VDCCs: voltage-dependent Ca^2+^ channels.

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
