# Peer review of "A Review of Endothelium-Dependent and -Independent Vasodilation Induced by Phytochemicals in Isolated Rat Aorta"

_animals, 2019, doi:10.3390/ani9090623_

Round 1
Reviewer 1 Report
This review by Knox et al focused on describing the effects of various plant extracts and phytochemicals on contraction and relaxation of isolated rat aorta. It is a good collection of literature information available on this topic although the English need some improvement. My main concern is the suitability of a review focusing on plant extracts and phytochemicals in the journal Animals. To better fit the scope of this journal, there should be more focus on detailing the advantages vs. disadvantages of using rat aorta in such experiments, what are the aspects that can be translated to other animals/human and where are the pitfalls and limitations. Maybe even briefly describe the process of rat aorta preparations, tips for successful experiments and troubleshooting. I feel such details focusing on rat anatomy/physiology will be more relatable to the scope and readers of this journal. However, this is by no mean a criticism of the scientific merit of this manuscript but more of an editorial decision to be made by editors. In addition, here are some of the other comments that I have:
1. Overall, English need to be improved. Certain sentences are too complicated and reads badly. For example, first sentence in the abstract and last sentence in the 2nd paragraph of introduction. I will point out some other language issues below but the list is not exhaustive. I am not a native speaker too, so my help is limited.
2. The authors also seems to ignore many nomenclature of scientific writing. Eg. scientific names are not italicized. Proper superscripts and subscripts not used (eg. charges on ions, EC50 etc.). Greek characters are often missing (eg. alpha-1 adrenegic in L59).
3. Last paragraph of Section 3 describe GPCR signal pathways but lack the link to vasodilation/contraction, perhaps the authors can add some information here.
4. Other specific errors I can spot:
a. L17: development of phytochemicals as drug or phytochemical discovery etc is probably better compared to “phytochemical’s development”
b. L28: “first cause of death” is not right, perhaps “top” or “number one” cause of death
c. L36: should be a acetylcholine-mediated or –induced vasodilation
d. L37: maybe main compound responsible
e. L63: maybe “other than” instead of “besides to”
f. L78: should be MeSH and need to expand the term in the first instance
g. L83: maybe evaluated for?
h. L96: GPCRs
i. L105: such as
j. L108 & L109: Ca2+-ATPase
k. L130 & L131: primarily
l. L154: alpha-1D
m. L205: L-arginine (not L-N)
n. L333: what does 10-3 to 1 mg/ml means?
o. L416: induces vasodilation by blocking
p. L563: involves instead of evolves
Author Response
Answer to reviewer 1:
R1: my main concern is the suitability of a review focusing on plant extracts and phytochemicals in the Journal Animals.
Answer: The revised version of the manuscript overcomes this criticism by highlighting the experimental model (isolated rat aorta) in which the vasoactive activities of the phytochemicals considered in this review have been evaluated.
R1: Maybe even briefly describe the process of rat aorta preparations, tips for successful experiments and troubleshooting.
Answer: The current version includes the critical aspects to obtain an isolated rat aortic preparation that allows collecting reliable bioactivity data.
R1: 1. Overall, English need to be improved.
Answer: In this version, English was thoroughly reviewed, both in grammar and style. We also consider the suggestions of the reviewer 1.
R1: 2. The authors also seem to ignore many nomenclatures of scientific writing. E.g. scientific names are not italicized. Proper superscripts and subscripts not used (e.g. charges on ions, EC50, etc). Greek charcters are often missing (e.g. alpha-1 adrenergic in L59).
Answer: In this version, all the aspects mentioned above by reviewer 1 were considered.
R1: 3. The last paragraph of Section 3 described GPCR signal pathways but lack the link to vasodilation/contraction, perhaps the authors can add some information here.
Answer: The curren version includes details of GPCR signal pathways related to vasodilation and contraction (please, see Section 2.2).
R1: 4. Other specific errors I can spot:
Answer: All the corrections and suggestion were incorporated in the new version.

Reviewer 2 Report
Animals
Manuscript Number: animals-545403
Knox et al.: A review of endothelium-dependent and -independent vasodilation induced by phytochemicals in rat isolated aorta
In the review, the authors focuses on studying the contribution of isolated rat aorta as an evaluation model for phytochemical’s development, especially those extracts, and metabolites produced by plants, and considered as medicinal plants or consumed as foods.
This review includes the valuable information for phytochemical’s study using isolated rat aorta, however, it must be necessary to revise the manuscript following points.
In the review, authors focused on isolated rat aorta rings. However, it is not clear that how they did not show the benefit to use isolated rat aorta rings. Authors should write about the benefits of using rats compared to other species.
2. Authors showed that the mechanism of contraction and relaxation on vascular smooth muscle in figure1. However, the figure does not correctly show the intracellular Ca2+ movement in contraction and relaxation. For example, receptor agonist, such as noradrenaline or phenylephrine not only inducing Ca 2+ release form SR, but also inducing Ca2+ influx via VDCC. Moreover, the figure shows increases of cGMP induce opening K+ channel, such as BK channel or Katp channel. Opening these K+ channels induce depolarization, leading to inhibit Ca2+ influx. Authors should correct figure1.
Animals
Manuscript Number: animals-545403
Knox et al.: A review of endothelium-dependent and -independent vasodilation induced by phytochemicals in rat isolated aorta
In the review, the authors focuses on studying the contribution of isolated rat aorta as an evaluation model for phytochemical’s development, especially those extracts, and metabolites produced by plants, and considered as medicinal plants or consumed as foods.
This review includes the valuable information for phytochemical’s study using isolated rat aorta, however, it must be necessary to revise the manuscript following points.
In the review, authors focused on isolated rat aorta rings. However, it is not clear that how they did not show the benefit to use isolated rat aorta rings. Authors should write about the benefits of using rats compared to other species.
2. Authors showed that the mechanism of contraction and relaxation on vascular smooth muscle in figure1. However, the figure does not correctly show the intracellular Ca2+ movement in contraction and relaxation. For example, receptor agonist, such as noradrenaline or phenylephrine not only inducing Ca 2+ release form SR, but also inducing Ca2+ influx via VDCC. Moreover, the figure shows increases of cGMP induce opening K+ channel, such as BK channel or Katp channel. Opening these K+ channels induce depolarization, leading to inhibit Ca2+ influx. Authors should correct figure1.
Animals
Manuscript Number: animals-545403
Knox et al.: A review of endothelium-dependent and -independent vasodilation induced by phytochemicals in rat isolated aorta
In the review, the authors focuses on studying the contribution of isolated rat aorta as an evaluation model for phytochemical’s development, especially those extracts, and metabolites produced by plants, and considered as medicinal plants or consumed as foods.
This review includes the valuable information for phytochemical’s study using isolated rat aorta, however, it must be necessary to revise the manuscript following points.
In the review, authors focused on isolated rat aorta rings. However, it is not clear that how they did not show the benefit to use isolated rat aorta rings. Authors should write about the benefits of using rats compared to other species.
2. Authors showed that the mechanism of contraction and relaxation on vascular smooth muscle in figure1. However, the figure does not correctly show the intracellular Ca2+ movement in contraction and relaxation. For example, receptor agonist, such as noradrenaline or phenylephrine not only inducing Ca 2+ release form SR, but also inducing Ca2+ influx via VDCC. Moreover, the figure shows increases of cGMP induce opening K+ channel, such as BK channel or Katp channel. Opening these K+ channels induce depolarization, leading to inhibit Ca2+ influx. Authors should correct figure1.
Author Response
R2: In the review, authors focused on isolated rat aorta rings. However, it is not clear that how they did not show the benefit to use isolated rat aorta rings. Authors should write about the benefits of using rats compared to other species.
Answer: The revised version of the review oversomes this criticism by emphasizing the advantages that offer the isolated rat aorta over other species.
R2: 2. Authors showed that the mechanism of contraction and relaxation on vascular smooth muscle in figure 1. However, the figure does not correctly show the intracellular Ca2+ movement in contraction and relaxation. For example, receptor agonist, such as noradrenaline or phenylephrine not only inducing Ca2+ release form SR, but also inducing Ca2+ influx via VDCC. Moreover, the figure shows increase of cGMP induce opening K+ channel, such as BK channel or Katp channel. Opening these K+ channels induce depolarization, leading to inhibit Ca2+ influx. Authors should correct figure 1.
Answer: The current version improved the Figure 1 by clarifying the signaling involved, both in the contraction and in the relaxation of vascular smoth muscle cells.

Round 2
Reviewer 1 Report
I would like to thank the authors for the revised manuscript. I would recommend the acceptance of this manuscript, however, I think the title should be modified to “isolated rat aorta” instead of rat isolated aorta.